# ETECADx: Ensemble Self-Attention Transformer Encoder for Breast Cancer Diagnosis Using Full-Field Digital X-ray Breast Images

**DOI:** 10.3390/diagnostics13010089

**Published:** 2022-12-28

**Authors:** Aymen M. Al-Hejri, Riyadh M. Al-Tam, Muneer Fazea, Archana Harsing Sable, Soojeong Lee, Mugahed A. Al-antari

**Affiliations:** 1School of Computational Sciences, Swami Ramanand Teerth Marathwada University, Nanded 431606, Maharashtra, India; 2Faculty of Administrative and Computer Sciences, University of Albaydha, Albaydha, Yemen; 3Department of Radiology, Al-Ma’amon Diagnostic Center, Sana’a, Yemen; 4Department of Radiology, School of Medicine, Ibb University of Medical Sciences, Ibb, Yemen; 5Department of Computer Engineering, College of Software and Convergence Technology, Daeyang AI Center, Sejong University, Seoul 05006, Republic of Korea; 6Department of Artificial Intelligence, College of Software and Convergence Technology, Daeyang AI Center, Sejong University, Seoul 05006, Republic of Korea

**Keywords:** breast cancer, hybrid CAD system, ensemble transfer learning, convolution neural network (CNN), transformer encoder, expert physician validation and verification

## Abstract

Early detection of breast cancer is an essential procedure to reduce the mortality rate among women. In this paper, a new AI-based computer-aided diagnosis (CAD) framework called ETECADx is proposed by fusing the benefits of both ensemble transfer learning of the convolutional neural networks as well as the self-attention mechanism of vision transformer encoder (ViT). The accurate and precious high-level deep features are generated via the backbone ensemble network, while the transformer encoder is used to diagnose the breast cancer probabilities in two approaches: *Approach A* (i.e., binary classification) and *Approach B* (i.e., multi-classification). To build the proposed CAD system, the benchmark public multi-class INbreast dataset is used. Meanwhile, private real breast cancer images are collected and annotated by expert radiologists to validate the prediction performance of the proposed ETECADx framework. The promising evaluation results are achieved using the INbreast mammograms with overall accuracies of 98.58% and 97.87% for the binary and multi-class approaches, respectively. Compared with the individual backbone networks, the proposed ensemble learning model improves the breast cancer prediction performance by 6.6% for binary and 4.6% for multi-class approaches. The proposed hybrid ETECADx shows further prediction improvement when the ViT-based ensemble backbone network is used by 8.1% and 6.2% for binary and multi-class diagnosis, respectively. For validation purposes using the real breast images, the proposed CAD system provides encouraging prediction accuracies of 97.16% for binary and 89.40% for multi-class approaches. The ETECADx has a capability to predict the breast lesions for a single mammogram in an average of 0.048 s. Such promising performance could be useful and helpful to assist the practical CAD framework applications providing a second supporting opinion of distinguishing various breast cancer malignancies.

## 1. Introduction

Recently, among all other cancers, breast cancer is second only to lung cancer as the most frequent type of cancer, increasing the mortality rate among women worldwide [1]. There are a variety of risk factors that can lead to the development of this cancer, including sex, family history, aging, gene mutations, estrogen, and so on. However, there is no guarantee that any of these factors can show accurate proof of breast cancer incidence [2]. Breast cancer is always a silent disease and appears suddenly if there is no routine check annually by the patients. Breast cancer has a multi-step process that involves various cell types, and it is still difficult to prevent globally. One of the best ways to avoid breast cancer is to diagnose it as soon as possible in the curable period of this malignant lesion [3,4]. Physicians and radiologists have advised using a variety of methods to find breast cancer, including digital mammography (DM), ultrasound (US), and magnetic resonance imaging (MRI). Breast cancer in females has surpassed lung cancer as the most often diagnosed malignancy in 2020, with over 2.3 million new cases and 685,000 deaths, counting a total rate of 11.7% among other cancers, followed by lung cancer with 11.4% [5,6]. Across the globe, the radiologists use DM modality to screen breast images in cranio-caudal (CC) and mediolateral oblique (MLO) views for breast cancer detection at the first stage, because of its ability to present scan tumors with a very low X-ray dose, cheaper test, and its availability in various hospitals around the world [7,8]. Due to a huge number of patients, or to get another opinion from the AI-based machine about the detected cancer, radiologists have been usually utilizing the computer-aided diagnostic (CAD) systems. Indeed, such emerging software systems demonstrate a good candidate as diagnostic tools that provide many benefits including lesion detection and segmentation even for tiny breast lesions [1]. In addition, the CAD system could perform analysis of a huge number of patients rapidly without any labor concentration and effort. The CAD system alongside mammogram images could provide the related information about the breast density, shape, and suspected anomalies including masses and calcifications, aiding in positive prognosis and high survival rate [1,9]. Furthermore, the available mammography modalities are in two types that provide 2D and 3D breast medical images; however, the radiologists encounter difficulties in distinguishing normal tissues from abnormal (benign or malignant) in the issued images. Therefore, it is crucial to develop an accurate classification CAD system for mammograms in order to minimize the likelihood of false positives and recall rates. Using the utility and power of the AI technology, the building CAD system could be easier, trustable, and reliable. This is due to the fact that the AI could derive a million or more deep high-level features at once without any user intervention [10]. Deep learning’s ability to handle enormous amounts of data has made it one of the most promising technologies in prior study. In image processing, voice recognition, and pattern recognition, the convolutional neural network (CNN) is the most used deep learning method. Their end-to-end technique predicts from the input images’ meaningful and relevant attributes. Since CNN techniques automatically extract features from the input image, they outperform the traditional approach and are, therefore, more widely used in the research community for image classification. Moreover, the usefulness of the ensemble learning approach in this classification problem has not yet been explored, despite the fact that most existing deep learning algorithms for medical disease detection rely on a single CNN model [11]. In order to improve classification accuracy based on binary or multi-classification, numerous studies have been presented in the literature based on a variety of deep learning models, such as VGG16, DenseNet201, ResNet50, transformers (ViT), ensemble, and so on [12,13,14,15,16,17,18,19,20,21,22,23,24,25]. In this regard, the objective of this study is to propose a novel CAD system (i.e., ETECADx) based on the hybridization strategy to process feature extraction from the input raw images and fuse both ensembles learning as well as the transformer-based approaches. The paper uses a transfer learning technique to assess the efficacy of six pre-trained deep learning models (namely, DenseNet201, VGG16, GoogleNet, InceptionResNetV2, Xception, and ResNet50 network) on digital X-ray mammograms. In the binary approach, DenseNet201, VGG16, and InceptionResNetV2 served as feature extractors to the transformer encoder network, while DenseNet201, VGG16, and Xception are used in the multi-class approach. The main contributions of this work are summarized as follows:A novel CAD system is designed to accurately and rapidly predict the breast cancer based on the hybrid scenario of ensemble transfer learning as well as the emerging transformer-based approach.Automatic image processing-based breast lesions region of interest (ROIs) extraction from the entire mammograms to perform more accurate trainable parameters during the fine-tuning process of the proposed AI models.A comprehensive experimental study over binary and multi-class approaches is conducted using the benchmark INbreast dataset in terms of selecting the proper AI models for the ensemble learning, achieving accurate and rapid prediction performance, and establishing reliable and feasible CAD system.An ablation study is performed to show the contribution of each ETECADx component in order to improve the diagnosis performance of the breast cancer.To validate and verify the proposed ETECADx framework, a private breast cancer real dataset is collected and annotated by three expert radiologists.

The remainder of this article is structured as follows: The related works are summarized in Section 2. The research techniques and materials are introduced in Section 3. The experimental results are shown in Section 4. Section 5 presents the discussion of experimental findings. Section 6 presents the suggestions for further research works and the conclusion findings.

## 2. Related Work

### 2.1. Deep Learning Based-CNN for Medical Breast Imaging

Breast cancer is the second disease-led cause of mortality in women, affecting around 12.5% worldwide [26]. Therefore, to alleviate such disease, early detection of breast lesions is very important to increase the survival rate. Thus, numerous studies based on deep learning techniques have been proposed to improve the detection rate of breast cancer, using mammogram images [1]. Many CNN-based CAD models that depended on the transfer learning technique were employed to recognize normal from abnormal images, aiming to enhance the classification accuracy, precision, and training and detecting speed. The Uniform Manifold Approximation and Projection (UMAP), principal component analysis (PCA), and univariate methods were used to reduce the feature dimensional in the proposed CNN-based CAD models [27,28,29]. In [18], the authors used the CNN model as feature extraction and PCA for the reduction of the feature dimensional. The proposed model successfully classified data from the two datasets, MIAS and INbreast, with 97.93% and 96.646% accuracy, respectively. The computational cost and execution time were decreased when PCA was used, but the classification performance did not change. Shen et al. [20] used single and four models to examine a collection of deep learning methods for the detection of breast cancer on mammography images from the CBIS-DDSM and INbreast datasets. For the CBIS-DDSM dataset, the specificity of the proposed model was 80.1%, the sensitivity was 86.1%, and the AUC was 91%. The best single model for INbreast was employed, which achieved a 95% AUC across all images for an independent test. However, AUC was improved to 98%, sensitivity to 86.7%, and the specificity to 96.1% after averaging results from four separate models. Finally, the authors in [3] used AlexNet, VGG, and GoogleNet for feature extraction, while the dimension of the extracted features was reduced using a univariate methods. The proposed model achieved 98.50% accuracy, 98.98% sensitivity, 98.99% specificity, and 98.06% precision. At another trend, the deep learning YOLO predictor was used to separate benign from malignant tissue on mammography images [22,30]. Al-Antari et al. [10] used regular feed-forward CNN, ResNet-50, and InceptionResNet-V2 for breast cancer classification, while the YOLO was utilized for automatically detecting breast tumors. An accuracy of 95.32% was achieved on the INbreast dataset when the InceptionResNet-V2 classifier was applied. Even though the YOLO detector accurately predicts input images, it might be challenging to find small clusters of micro-calcification objects [30]. As the authors claimed, the micro-calcification is another trend and needs more investigation. Hamed et al. [24,29] utilized the YOLO classifier to classify benign and malignant mammogram images that were collected from the INbreast dataset, reaching an overall accuracy of 89.5% in [24] and 95% in [29]. Aly et al. [22] presented a study to use the ResNet and Inception models as a feature extraction methods and the YOLOv3 model as a classifier to detect benign and cancer masses. Such a model was successful in detecting 89.4% of the masses, with a precision of 94.2% and 84.6% for benign and malignant masses, respectively.

### 2.2. Ensemble Learning as a Backbone for Accurate Deep Feature Generation

An ensemble approach in machine learning is a technique that combines a number of single models in order to address a specific issue [31]. A few studies used an ensemble approach to distinguish benign from abnormal images for the breast cancer research domain [32]. Indeed, all of these studies concluded that the ensemble strategy has more capability than the individual AI model to achieve a higher prediction accuracy. This is because the ensemble learning could combine the best qualities of various contributions from multiple classifiers at once [32]. From a practical overview, it seems initially a more complex approach and costs more time for the computation, especially for the training process since various AI models are involved at the same time and fine-tune their weights simultaneously. To minimize such worries, domain researchers usually try to optimize and fine-tune the weights of the AI models individually using the same datasets in a unique environmental execution. Then, the trained models are fused all together using a single backend database structure for performing the testing and validation procedure. Such remedies could support and help the researchers as well as the real applications in the business company production line. Currently, many companies around the world try to involve AI technology for providing accurate and rapid smart solutions, especially in the medical domain. During a pandemic or epidemic, smart solutions are always required to support the limited medical staff in the hospital, healthcare centers, and so on. The authors in [32] presented a study using the Inception v4 Ensemble model with the fuzzy rank-based Gompertz function, in which the accuracy obtained was 99.32%. Chakravarthy et al. [33] developed an improved crow search-optimized extreme learning machine (ICSELM), and such a model attained an accuracy of 98.26% for the INbreast dataset. Thuy et al. [34] improved classification performance using a hybrid deep learning model that combined the VGG19 and VGG16 models with a generative adversarial network (GAN) and achieved 98.1% accuracy. On the other hand, Savelli, Benedetta, et al. [35] proposed a multi-context ensemble of convolutional neural networks for detecting small tumors (microcalcification) using the mammograms of the INbreast dataset. Such a model achieved a 36.25% free receiver operating characteristic (FROC) score based on the INbreast dataset. Furthermore, Sahu, Yatendra, et al. [36] suggested an ensemble technique by using ReNet18 and support vector machines (SVM) as a CAD system, where the pretrained ReNet18 model is used as a feature extraction and SVM to classify breast cancer lesions based on the BreakHis dataset. The proposed model provides superior accuracy at 200× magnification of 92.6%. The 100× magnification factor yields the maximum specificity and precision, which are 93.1% and 86.5%, respectively [11]. Last but not least, Samee, Nagwan Abdel, et al. [37] used a hybrid technique based on logistic regression (LR) and principal components analysis (PCA), targeting the important components involved in the classification process. Using the INbreast and mini-MAIS datasets, the suggested CAD system could reach the best performance accuracies of 98.60% and 98.80%, respectively.

### 2.3. Vision Transformer-Based Medical Image Classification

Recently, the vision transformer (ViT), based on a self-attention mechanism, has demonstrated considerable promise in image classification [38]. The ViT principle was involved to classify images in which the input images were breaking into patches with a fixed size and which are then connected together linearly to form a vector and processed by a traditional converter encoder [39]. A few studies have been suggested based on the ViT principle, for example, the authors in [38] used a ViT model for classifying breast cancer using ultrasound images. In [40], the local features of the input images were extracted using a CNN module, while a ViT module was used to improve the global features for identifying different regions in the input images. The proposed hybrid model attained values of 90.73%, 90.77%, 85.58%, and 90.73% in terms of recall, precision, specificity, and F1 score, respectively. In [41], the ViT-based semi-supervised learning model using ultrasound and histopathology datasets was also used for the classification of breast cancer. The suggested model outperformed CNN models of DenseNet201, ResNet101, and VGG19 by achieving 96.29% precision, a 96.15% F1-score, and 95.29% accuracy. Chen et al. [42] suggested using the local and global transformer blocks to model within four mammograms taken from both views for each side. The four images were then combined into a single sequence global transformer and passed into the MLP head for classification. They achieved AUC of 0.784. Al-Tam et al. [2] proposed a new hybrid model that involved a transformer encoder with multiple layer perceptron (MLP) for classification based on the high-level deep features extracted via ResNet50. Their proposed model outperformed against others individual classification models of ResNet50, VGG16, and Custom CNN. The suggested CAD system was built and tested using two datasets, CBIS-DDSM and DDSM. The evaluation findings for the proposed hybrid CAD system reached overall accuracies for the binary and multi-class predictions of 100% and 95.80%, respectively. In [43], He et al. used a Deconv-Transformer (DecT) model that includes a color deconvolution as convolution layers to classify breast cancer based on histopathological images collected from the BreakHis dataset. Their proposed model achieved an average accuracy of 93.02% and an F1-score of 93.89%. The ensemble of the Swin transformer (SwinT) model was proposed to differentiate benign from malignant cancer in the histopathology breast images from the BreakHis dataset [44]. The model investigated eight different subtypes for each of the two classes, and it demonstrated an average test accuracy of 96.0% for the eight-class and 99.6% for the binary classification.

## 3. Material and Methodology

### 3.1. The Proposed AI-Based ETECADx Framework

To achieve the goal of breast cancer malignancy diagnosis in an accurate and rapid manner, the ETECADx framework is proposed. We design and build the proposed AI framework considering the practical CAD application perspective starting with the medical data collection into the final prediction scores of the potential breast cancer malignancy level: benign or malignant. Figure 1 shows the consecutive processing stages of the proposed ETECADx, including medical benchmark data collection, preprocessing, constructing the desired AI model based on recent technologies, and fine-tuning, validating, and evaluating the prediction performance. We adopt and fine-tune the AI framework using the benchmark INbreast dataset since it has proven class-wise labels (i.e., normal, benign, and malignant) as well as the contour of the breast lesions. After that, private medical breast images are carefully collected and annotated for further validation and verification. Generally, the first step to build a proposed CAD system is to design the end-to-end scenario considering, in some sense the practical abstract pipeline view. While breast cancer is one of the most silent diseases, the patient should regularly visit the healthcare center and consult the specialist at least twice a year. Such health recommendation is announced by the United States National Center Institute (NCI) [45]. After consulting with the specialist radiologist or physician, the mammogram test might be assigned. In this case, the patient’s breasts will be X-ray scanned using the standard tool of the mammographic device providing at least two MLO and CC views for each breast. The scanned breast images are stored in a DICOM format for investigating and providing the proper recommendation to the patient. From the R&D perspective, the data DICOM format must be converted into a readable format (i.e., png, jpg, jpeg, tiff, etc.) for the PC machines. Meanwhile, we request the expert radiologist for a further annotation process to accurately determine the class-wise classification label, lesion boundary segmentation, and detection bounding box. Once the breast images and their annotations are available, the AI researchers and developers are able to launch and build the proper diagnosis CAD framework. The preprocessing step is required to remove unwanted details, and improve the image quality, image dimension adjusting, and intensity normalization. In the medical research domain, it is proven that such preprocessing process could significantly improve diagnosis accuracy [46]. One of the most important prior steps is to extract the potential lesion ROIs or patches. This is to consciously optimize the proper input image size, allowing the AI model to fine-tune its trainable parameters based on specific and accurate malignancy areas. In this way, the AI models could enrich their knowledge better than if they train using the whole input images. This is because the lesion size inside the medical images is mostly tiny comparing the whole image size. Such critical issue to building a more feasible and reliable CAD system was investigated in detail in our previous studies [10]. Here, we extend our work to further check the possibility of the most recent AI strategies to improve prediction performance. Thus, we target both new strategies of ensemble transfer learning and the ViT based on the self-attention mechanism. Such a hybridization strategy encouraged us to take a further step for more improvement to enrich the research domain and might support the real practical sectors, such as companies, research health centers, and so on.

### 3.2. Dataset: Digital X-ray Mammograms

To train or fine-tune the AI models, the benchmark public multi-class INbreast dataset [47] is used. In addition, the private real breast cancer images are collected and annotated by expert radiologists to only validate the proposed AI-based ETECADx framework. The details description of both datasets are addressed in the following sections.

#### 3.2.1. INbreast Public Dataset

The breast images of the INbreast dataset were collected at a breast center located in a university hospital (Centro Hospital de So Joo [CHSJ], Breast Center, Porto, Portugal) with the approval of the Portuguese National Committee for Data Protection and the Hospital’s Ethics Committee from April 2008 to July 2010 [47]. INbreast comprises a total of 410 images collected from 115 patients, including 360 images from 90 women affected in both breasts (four images for left and right sides with both views CC and MLO). A total of 25 patients underwent mastectomy (one side only with two views). Multiple forms of lesions, such as masses, calcifications, and deformities, are included in the dataset. Specialists produced contours of lesions in XML format with DICOM images [47]. There are 107 breast lesions analyzed and scored by the BI-RAD system where 36 mammograms with BI-RAD of 2, and 3 are considered to represent benign mass cases. Whereas, 71 mammograms with BI-RAD scores of 4, 5, and 6 are considered to represent the malignant mass cases. For normal class, 67 cases were collected. Figure 2 displays breast mammograms for three distinct patients collected from the INbreast dataset (i.e., Dataset1), where normal, benign, and malignant examples are shown in Figure 2a, Figure 2b and Figure 2c, respectively.

#### 3.2.2. Private Real Breast Images

The real breast mammograms are used only to validate the proposed ETECADx framework. All mammograms are collected from Al-Ma’amon diagnostic center in Sana’a, Republic of Yemen from March to August 2022 under the supervision of two local expert radiologists: Amal Abdulrahaman Bafagih and Muneer Abdulwasea Fazea. Both of these experts are officially working in the Al-Ma’amon diagnostic center to consult patients regarding breast scanning, investigation, diagnosing, and recommendations. To scan the patient breasts, the standard tool of the mammographic device is used: a Senographe 800T High-Frequency X-Ray Generator delivering constant voltage, GE, Florida USA (3D mammography machine). The detailed clinical English report of each case is well prepared by those expert radiologists (local experts), to explain the case condition, diagnosis annotation, tumor size, location, and so on. The experts always follow the BIRAD scale to score the breast cancer as a benign or malignant. For further data investigation in terms of accurate breast cancer annotation (benign or malignant) and the lesion localization on the mammograms, we invite the international expert radiologist Dr. Rajesh Kamalkishor Agrawal who is the director of the Nanded Life-Line Private Clinic, India. He has deep work experience as a radiologist in a breast cancer clinic for around 30 years. Due to the lack of supported ultrasound scans for some cases, the international expert could not judge the exact cancer findings. Thus, the number of benign and malignant cases are decreased to 25 and 101 cases, respectively. To use such a dataset in our study, we carefully check and pick up the class-wise annotation or label (i.e., normal, benign, malignant) for each mammogram based on the recommendation from both local and international experts. The annotation agreement ratios among local and external expert radiologists are calculated based on the kappa agreement factor to be 80.16%, 50.0%, and 100% for malignant, benign, and normal cases, respectively. Therefore, we carefully choose only the accurate label images where both local and external evaluators agree on their labels (i.e., Dataset2-B in Table 1). Other cases with annotation conflicts are excluded from this study. Table 1 shows the data distribution over each class: normal, benign, and malignant. All breast images are collected for both CC and MLO views, except for six cases, where the tumor area is located in the upper quarter towards the axillary side, which makes it impossible to appear on the CC views. Figure 3 shows examples of the breast mammograms in normal, benign, and malignant conditions.

### 3.3. Medical Data Preprocessing

The medial data preprocessing is always needed to well prepare the trainable breast images, remove unwanted or useless information for the AI classifiers, improve the image spatial resolution and quality, and normalize and resize the pixel intensities to fit in a single gray scale range for all images [46]. First, the breast images are converted from the DICOM image format into the “png” format according to the unique patient ID and BI-RADS classification scores, 0, 1, 2, 3, 4, 5. The “0” score reflects the normal cases, “1” and “2” reflect the beaning cases, and the scores of “3” to “5” explain the malignant cases. In addition, the breast lesion accurate contour of each breast tumor is precisely determined by expert radiologists for training AI models based only on these regions instead of using the whole mammograms. Such medical sensitive information is carefully prepared by expert radiologists and is publicly available. Second, all mammograms are read as a whole full-size image without downscaling to keep images in high resolution before the image patch extraction process. Third, the patch images are extracted to include only the ROIs of the breast lesions ignoring other background information. This is the most important preprocessing step, to enable the AI models to fine-tune their weights based on the accurate malignancy regions instead of using the whole image [20], as it is known that the tumor size in the medical images is very small compared with the whole image size. So, if the AI model is trained on the whole image where the majority of pixels are not related to the tumor itself, the optimized weight parameters will be weak and not be enough to achieve an impressive overall diagnosis accuracy. In this study, we extract the tumor ROIs based on the available ground truth (GT) annotation mask in the XML file per mammogram. Furthermore, all extracted image patches are resized into 512 × 512 pixels. This is to enable the AI models to train on the same image characteristics and reduce the GPU processing time, especially with a huge dataset; this is mandatory for deep learning models. To do this, we use the OpenCV *bitwise_AND* image processing technique where the morphological operations match both the contour binary mask with its associated original images [48]. Figure 4 shows an example of the extracted breast lesion ROI from the whole mammogram, Figure 4a the original image with lesion couture, Figure 4b the output of the OpenCV *bitwise_AND* image processing strategy, and Figure 4c extracted ROI image. The detailed scenario in how the ROI is accurately extracted is explained in the next section.

### 3.4. Patch Image Extraction

As stated previously, the size of a whole breast image is extremely huge and the unwanted area must be removed. In this research, image processing based on the function of *bitwise_AND* operation is used to extract or segment lesion ROIs based on the experts’ annotation. The OpenCv functions of cv2.threshold, cv2.findContours, and cv2.boundingRect are used to perform the lesion segmentation as follows. First, the cv2.threshold function is applied to the input image via the lower and upper threshold limits. Here, we use the binary Otsu thresholding approach to pass as an additional flag where the threshold value can be chosen at random [49]. Second, the cv2.findContours function is used to find the breast lesion contour with the following inputs: first step segmented image, cv2.RETR EXTERNAL, and cv2.CHAIN APPROX SIMPLE. The RETR EXTERNAL is just used to retrieve the extreme outer contours, while the CHAIN APPROX SIMPLE returns the endpoints that are necessary for drawing the contour over the input image. Finally, the largest contour is detected by the max value operator from the contour area using cv2.contourArea function. The extracted bounding rectangle with new dimensions of x, y, width, and height are extracted using the function of cv2.boundingRect as shown in Figure 4c. The final segmented and cropped ROIs are used as input patch images to execute our experiments for this study. This method is used because it saves memory without sacrificing output quality. Algorithm 1 shows the pseudo-code for preprocessing procedure.
**Algorithm 1:** Patch Image Extraction via Image Preprocessing Approach**Start**:Input: Original image with its mask**Step 1: load data**ImageMaskcv← image; {read DICOM image format}← mask; {load mask from XML file}← openCV2; {python library for computer vision task}**Step 2: Apply***bitwise_AND*Image ← cv2.bitwise_and (Original Image, Mask, cv2.COLOR_BGR2GRAY)**Step 2: Binary thresholding**   1.BEGIN   2.READ bitwise_And image, s(x,y) where x and y denotes the pixel coordinates   3.def crop (bitwise image) {   4.threshold value, t = 0   5.maximum value, m = 255   6.thresh = cv2.threshold (bitwise_images, t, m, cv2.THRESH_OTSU + cv2.THRESH_BINARY) [1]   7.cnts, _ = cv2.findContours (thresh, cv2.RETR_EXTERNAL, cv2.CHAIN_APPROX_SIMPLE)   8.cnt = max (cnts, key = cv2.contourArea x, y, w, h = cv2.boundingRect (cnt)   9.return bitwise image [y:y + h, x:x + w]**END**

For normal cases, the mammograms are segmented and cropped into multiple 512 × 512 pixel patches. However, since the breast image is on one side of the image and the opposite side has a black background, we do not require the majority of clips with a black background. Since the image size is 512 × 512 pixels, images are read pixel by pixel and black pixels are counted and removed if they exceed 25% of the whole image size. Algorithm 2 demonstrates the segmentation and cropping of normal images.
**Algorithm 2:** Patch Image Extraction for Normal Cases1. **START:**2.  def tile (image,input_path,output_path,dim) {3.  Declare name, ext, image, w, h4.  image = Image.open (os.path(image))5.  w, h = Image.size6.  grid = product (range(0, h-h%d, d), range(0, w-w%d, d))7.  FOR i, j in grid:8.   box = (j, i, j + d, i + dim)9.   out = os.path.join(dir_out, f’{name}_{i}_{j}{ext}’)10.   img.crop(box).save(out)11.  ENDFOR12. END**Ignore the images with the majority in black**1. START2. zero = skimage.io.imread(fname = image_name)3. declare black_counter = 04.  for i in range (image.size [1]):5.   for j in range (image.size [0]): 6.    if zero[i,j].any() <=0:7.     black_counter = black_counter + 18.  if black_counter<=image.size/4:9.   shutil. copy2 (image_name, new_path)**END**

### 3.5. Data Preparation for Training, Validation, and Testing

The data are split into binary classification and multiclass recognition approaches. A total of 70%, 20%, and 10% of all breast images from each class are randomly divided into training, testing, and validation sets, respectively, for both approaches. The distribution of data for the first approach, in which they were classified as normal and abnormal, is depicted in Table 2. After augmentation procedures, a number of normal images were taken to balance the abnormal images. For the second approach, according to total number of mass images, Table 3 shows the distribution of multi-classes. In the private real dataset, the data splitting in binary approach is 101 images for malignant, and 110 images for normal. Whereas the multi-class approach is 25 images for benign, 101 images for malignant, and 110 images for normal.

### 3.6. Training Data Augmentation

A large enough dataset is necessary for the training of deep learning-based models. When working with medical images, for example, such a dataset is often not readily available; as a result, data augmentation has become a common technique for solving this problem [50,51]. In the INbreast dataset, the images for each class are not balanced. After splitting the data, the training data is made up of 25 benign images and 49 malignant images. For normal cases, we use, in both approaches, all 597 patches that generated from a whole image segmentation process. The benign training set are flipped vertically to 50, then all benign and malignant training sets are added by rotating 45, 90, 135, 180, 225, 270, and 315. In a binary approach, the total of training set is 1010 (418 normal and 592 abnormal), but in a multi-class approach, the total is 1210 (i.e., 418 normal, 400 benign, and 392 malignant). Table 2 and Table 3 present the original and augmented dataset distribution per class for *Approach A (binary classification)* and *Approach B (multi-class classification)*, respectively. The normal patch images are generated from the original normal mammograms without augmentation, while the augmentation is done for abnormal cases to enlarge the number of instances and balance the normal and abnormal cases. This is to avoid any bias due to the majority samples of any class during the training and optimization process of the trainable parameters. The balanced dataset per class is helpful to improve the diagnosis performance of the AI-based models [10].

### 3.7. Ensemble Transfer Learning

Recently, the concept of ensemble learning has been employed in place of a single deep learning model in order to learn several deep learning models [11,52]. Therefore, the prediction process is finally done by merging multiple different models. This makes it possible to take advantage of more useful information from the different classifiers and obtain more accurate classification results. In previous studies, most of the deep learning techniques for breast cancer prediction relied on a single convolutional network, as the applications of ensemble and transformer learning for the early detection of breast cancer are still in the infant stages, and some studies apply to histopathological images. In this paper, the ensemble learning method was built by selecting the best set of pre-trained deep learning models as a main structure for the extraction of features. As illustrated in Figure 5, this paper’s ensemble technique concatenates pre-trained deep learning models for feature extraction.

For the proposed ensemble learning model, we combined the deep learning features of DenseNet201, VGG16, and InceptionResNetV2 for the binary approach, while DenseNet201, VGG16, and Xception are combined for the multi-class approach. Each model’s first top layer, its classification layer, is eliminated, and its last block convolution layer is mined for its deep features. As shown in Figure 1, the proposed hybrid AI model is built based on fusing or ensembling the high-level features from three CNN-based backbone networks. For each CNN model, the classification dense layers are eliminated first. Thus, the final deep features are extracted from the top layers of DenseNet’s, VGG16, and Xception, as represented by (None, 16, 16, 1920), (None, 16, 16, 512), and (None, 16, 16, 2048), respectively. To average the high-level deep features from each model and convert them to a vector feature, global average pooling (GAP) is applied. After that, the high-level deep feature vectors from three CNN models are fused or concatenated together to produce the feature vector space. Finally, the feature vector is embedded and passed into this prediction stage of ViT, as discussed in the Section 3.8.

To build and select the proper candidates of our study, six CNN-based deep learning models are randomly selected due to their excellent reputation in the computer vison research domain for image classification. More explanation of these models is described as follows:*AI-based VGG16:* VGG16 is one of the popular CNN pre-trained models used for classification tasks. In this work, the classification layers from the pre-trained VGG16 model that was trained on the ImageNet dataset were deleted before this model is applied. Therefore, a new classification layer is added for binary and multi-classification, in which the highest performance is achieved. A conventional layer with 1024 neurons, batch normalization, and dropout (50% of dropout rate) layers are added, respectively. The fine-tune used for binary and multi-classification is 17, in which all layers starting from the layer number 17 to the classification layers are trainable, while the rest are untrainable.*AI-based ResNet:* ResNet is a deep convolutional neural network with 50 layers that has been used for image identification applications. ResNet50 was trained using the ImageNet dataset for classifying almost 1000 classes, similar to the pre-trained VGG16 model, and the classification layers are also deleted. We add the same layers for binary and multiple classifications, just like in the AI-based VGG16 in the above section. Two fine-tuned configurations are applied for the ResNet50 model: 143 is used for binary classification and 123 is applied for multi-classification, since they recorded the highest performance.*AI-based DenseNet201:* All layers in the DenseNet model are linked together in a feedforward approach. Each successive layer receives its own feature maps and also receives inputs from all preceding levels [53]. The structure is finished with the addition of global average pooling, one fully connected layer, and a SoftMax layer. All training is done on the fine-tune layer, with a fine-tune value of 481. This is utilized for both binary and multi-classification approaches.*AI-based GoogleNet:* The purpose of developing this model was to solve the issue of overfitting whilst also delving further into the network layer [54]. Out of a total of 311 layers structured into convolution layers, max-pooling layers with nine linearly stacked Inception modules, 252 are trained using fine tuning. Then, global pooling is added with one fully connected layer and output layer.*AI-based InceptionResNetV2:* The network architecture is similar to that of InceptionResNetV1, but the stem is based on InceptionV4 [55]. On the far left of each module is a shortcut link. For better classification results, it blends inception architecture with residual connections. The convolutional operation in the inception module needs to take the same input and output for the inception convolutional operation.*AI-based Xception:* This is a stack of linearly connected convolution layers that can be separated by their depth. It is made up of 36 convolutional layers that are organized into 14 modules. Each module has linear residual connections around it, which serve as the network’s backbone for extracting features [56].

### 3.8. The Proposed Hybrid AI Model

The proposed Hybrid AI model is designed and constructed based on the promising advantages of recently developed ensemble transfer learning AI approaches as well as the ViT as shown in Figure 1. By combining different CNN-based models, ensemble learning is used as the backbone network to provide high-level deep features. In order to strongly derive more accurate features instead of employing a single model, the fusing technique has recently been introduced in computer vision image classification [57]. The vision transformer is primarily utilized due to its ability to diagnose objects more precisely based on the precious, deeply derived sensible features, whilst the self-attention features are utilized due to their high performance and reduced demand for vision-specific inductive bias [58]. The transformer is a deep learning-based method that uses self-attention to apply various weights for calculating the significance of each input data in an encoder–decoder configuration [41]. The CNN models merely examine the association between spatially adjacent pixels in the receptive area established by the filter size [11], and thus distant pixels cannot be handle by such models. Therefore, to address this issue, a new trend was implemented based on the attention mechanism. The attention technique depends on determining and processing the most informative parts of the data (images), in which the redundant parts will be discarded; thereby, false negative results well be reduced. In this paper, the vision transformer via encoder was adopted and fine-tuned. A self-attention network, a multi-linear perceptron block, and a classification layer make up the proposed transformer encoder. The self-attention mechanism is responsible for connecting various locations within the same input data, creating a single input sequence [41]. Figure 1 shows how the ViT model linearly concatenates 16 × 16 2D patches of the input image into 1D vectors, which are then fed into a transformer encoder of multi-head self-attention (MSA) and MLP blocks. To determine the connection between each patch and all other patches in a single input sequence, the MSA uses a scaled dot-product form of attention, as shown in Equation (1):(1)$$Attention\left(Q,K,V\right)=Softmax\;\left(\frac{Qk^{t}}{\sqrt{d_{k}}}\;\right)v,$$
where *Q* means query vector, *V* is a value dimensional vector, and *K* refers to the key vector. The $d_{k}$ represents the variance of the product $Qk^{t}$, which has a zero mean. In addition, normalizing the product by dividing it by the standard deviation $\sqrt{d_{k}}$. The SoftMax function converts the scaled dot-product into an attention score. This mechanism is the key of the transformer model for offering parallel attention to comprehending the input image’s overall content. The model can respond to input from numerous representation subspaces at various locations simultaneously due to the multi-head attention. The multi-head attention linearly extends the queries, keys, and values *h* times using a variety of learnt linear projections, and can be calculated by Equation (2).
(2)$$\begin{array}{c} \mathrm{MultiHead}\left(Q,K,V\right)=\mathrm{Concat}\left({\mathrm{head}}_{1},\ldots ,{\mathrm{head}}_{h}\right)\;W^{o}\\ \mathrm{where}\;\mathrm{headi}\;=\;\mathrm{Attention}\left(QW_{i}^{Q},KW_{i}^{K},VW_{i}^{V}\right), \end{array}$$
where the projections are parameter matrices $W_{i}^{Q}\in R^{d_{model}\;x\;d_{k}},{\;W}_{i}^{K}\in R^{d_{model}\;x\;d_{k}}$, $W_{i}^{V}\in R^{d_{model}\;x\;d_{v}}$ and $W^{o}\in R^{hd_{v}\;x\;d_{model}}$. On the other hand, the MLP block consisted of a non-linear layer of Gaussian error linear unit (GELU) with 1024 neurons and batch normalization, with a dropping rate of 50% across all dropout layers.

### 3.9. Experimental Setup

End-to-end training is used for the proposed AI hybrid model. In this study, we employed a learning rate of 0.001, an Adam optimizer with a clip value of 0.2, and a patience of 30 when conducting our training and in stopping callback early. We train all AI models using 100 epochs to fine-tune the hyper-parameters of the models. In the encoder section, an image size of 512, a patch size of 2, an input size of 20, a drop rate for all layers is 0.01, and 8 heads. Meanwhile, to show the dimension by which high-dimensional vectors are converted to low-dimensional vectors without loss, the embed dim and num_mlp are empirically optimized to be 64 and 256, respectively.

### 3.10. Evaluation Strategy

The evaluation matrices of Accuracy (Acc.), Specificity (SPE), Sensitivity (SEN), F1-score, Matthews correlation coefficient (MCC), Cohen’s kappa coefficient, and the receiver operating characteristic (ROC) curve are used to measure the prediction performance of the proposed AI framework, as in our previous works [59]. The mathematical definitions of all these metrics are defined as:(3)$$\mathrm{Accuracy}\;\left(\mathrm{ACC}\right)=\frac{\mathrm{TP}+\mathrm{TN}}{\mathrm{TP}+\mathrm{TN}+\mathrm{FP}+\mathrm{FN}}.$$
(4)$$\mathrm{Specificity}\;\left(\mathrm{SPE}\right)=\frac{\mathrm{TN}}{\mathrm{TN}+\mathrm{FP}}.$$
(5)$$\mathrm{Sensitivity}\;\left(\mathrm{SEN}\right)=\frac{\mathrm{TP}}{\mathrm{TP}+\mathrm{FN}}.$$
(6)$$F1\text{-}\mathrm{Score}=\frac{2\mathrm{Precision}\times \mathrm{Sensitivity}}{\mathrm{Precision}+\mathrm{Sensitivity}}.$$
(7)$$\mathrm{MCC}=\frac{\mathrm{TP}.\mathrm{TN}-\mathrm{FP}.\mathrm{FN}}{\sqrt{\left(\mathrm{TP}+\mathrm{FP}\right)\left(\mathrm{TP}+\mathrm{FN}\right)\left(\mathrm{TN}+\mathrm{FP}\right)\left(\mathrm{TN}+\mathrm{FN}\right)}}$$
(8)$$\mathrm{kappa}=\frac{P_{o}-P_{e}}{1-P_{e}}\;,\;\mathrm{where}\;P_{o}=\frac{\mathrm{TP}+\mathrm{TN}}{Total},\;\mathrm{and}\;P_{e}=\frac{\left(\mathrm{TP}+\mathrm{FP}\right)\left(\mathrm{TP}+\mathrm{FN}\right)+\left(\mathrm{TN}+\mathrm{FP}\right)\left(\mathrm{TN}+\mathrm{FN}\right)}{Total^{2}}\;$$

The binary and multi-class confusion metrics are used to derive the parameters of true positive (TP), true negative (TN), false positive (FP), and false negative (FN). Meanwhile, the area under the receiver operator characteristic (ROC) curve (AUC) was used to evaluate a classifier’s capacity to distinguish between classes. We solve the binary class problem approach by employing the “roc_auc_score” built-in function included in the Python Sklearn module [60]. Similar to our prior work [9], we use a one-class-versus-others technique to construct ROC curves with their AUC values for the multi-classification approach. After that, the estimated mean AUC values are reported. Due to the imbalanced testing set per class, the weighted evaluation metrics strategy is used for both A and B approaches.

### 3.11. Execution Environment

An MSI GS66 laptop with the following specifications is used to carry out the experiment: INTEL CORE I 7 11TH GENERATION (11800H), 32 GB RAM with 2 TB SSD NVME and RTX 3080 (16 GB) GRAPHICS CARD. Python 3.10 running on Windows 11 along with the Keras and TensorFlow backend libraries were utilized in the conduct of the experiments that are analyzed in this study.

## 4. Experimental Results

The evaluation prediction results are presented in this section for Approach A (binary classification) and Approach B (multi-class classification) as shown in Figure 6. For each approach, the results of individual AI-based CNN model are presented first. Then, the comparison results between the ensemble learning model as well as the proposed ETECADx are consecutively demonstrated.

### 4.1. Approach A: Binary Classification Results

#### 4.1.1. Individual Pre-Trained Deep Learning Models

For this study, the state-of-the-art deep learning individual models of DenseNet201, VGG16, GoogleNet, InceptionResNetV2, Xception, and ResNet50 are selected and investigated in order to choose the best combination of the ensemble backbone network of the proposed ETECADx framework. We optimize and fine-tune these models individually using the INbreast dataset. Their trainable parameters are experimentally selected based on the strategy of multi-trail errors [61]. The evaluation prediction results of this approach for each AI model are presented in Table 4. The best classification results are clearly recorded for the DenseNet201 and VGG16 in terms of all evaluation metrics where the overall accuracy and F1-score are achieved with 95.74% and 95.66%, respectively. The second better prediction performance is achieved using the InceptionResNetV2 recording the overall accuracy and F1-score of 93.62% and 93.60%, respectively. On the other hand, the ResNet50 achieved the lowest performance compared with other AI models. As a conclusion of this study, the proper candidates that could be used to build the ensemble AI backbone model are DenseNet201, VGG16, and InceptionResNetV2.

Based on the confusion matrix, the DenseNet201 and VGG16 achieved the best prediction performance with only six cases wrongly classified (four in abnormal class and two in normal class). The InceptionResNetV2 misclassified five abnormal cases as normal. The Xception achieved good classification behavior to accurately distinguish the normal cases, but it is the worst to predict the abnormal cases where the false negative is very high at thirteen cases. Figure 7 shows examples of the confusion matrices of the three AI-based CNN models: DenseNet201, InceptionResNetV2, and Xception.

#### 4.1.2. Ensemble Learning vs. the Proposed ETECADx

Based on the results of the individual pre-trained AI models, DenseNet201, VGG16, and InceptionResNetV2 are used to build the ensemble backbone network for the binary approach of the proposed ETECADx system. In this section, we perform the evaluation performance in two steps. First, the ensemble learning model is constructed, trained, and evaluated separately without using the ViT. Second, we construct the proposed hybrid model by allowing the ViT to perform the final prediction performance based on the ensemble high-level deep features that are generated by fusing three different CNN models: DenseNet201, VGG16, and InceptionResNetV2. The evaluation performance results of the breast cancer diagnosis using the ensemble learning and the proposed hybrid ETECADx are summarized in Table 5. It is clearly shown that the proposed AI ETECADx model could achieve the superior diagnosis evaluation results when the ensemble learning model is used. Figure 8 presents the confusion matrices for the proposed ensemble learning and ETECADX AI models for the binary classification approach.

### 4.2. Approach B: Multi-Classification Results

#### 4.2.1. Individual Pre-Trained Deep Learning Models

For the multi-class approach, the same individual deep learning models are selected and investigated for building the proposed models: ensemble backbone as well as the proposed ETECADx. For *Approach B*, the individual models show their capabilities to distinguish the multi-classes as shown in Table 6. The high evaluation prediction of accuracy is achieved via VGG16 and Xception with 95.74%, and 95.04%, respectively. Whereas, the GoogleNet achieves the lowest classification accuracy of 85.11%. Thus, the best candidates for building the ensemble learning models are VGG16, Xception, and DenseNet201.

The examples of the confusion matrix of the best three individual models are depicted in Figure 9. The VGG16 could achieve the superior classification performance where only one benign case is misclassified as a normal. These three AI models get confused in classifying three benign cases as a normal, achieving similar false negative ratios. Based on this behavior, we build the backbone network using the best individual CNN models: VGG16, Xception, and DenseNet201.

#### 4.2.2. Ensemble Learning vs. the Proposed ETECADx

Similarly, the proposed ETECADx framework is evaluated twice. First, the ensemble learning model is considered to provide the final prediction of breast cancer. Second, the proposed hybrid ensemble and ViT model is used to perform the final breast cancer prediction. The evaluation performance results of both cases are summarized in Table 7. For the multi-classification approach, the proposed ETECADx using the hybrid ensemble learning as a backbone and ViT as a predictor could achieve the superior evaluation performance with an accuracy of 97.87% compared with 96.45% for the ensemble model alone. Although the prediction performance of the ensemble AI model is better than individual models, the use of ViT is recommended to increase the prediction performance of breast cancer. The evaluation result is also presented in terms of confusion matrices for the ensemble learning model as well as the proposed hybrid ETECADx. Figure 10 presents the confusion matrices for the proposed ensemble learning and ETECADX AI models in the multi-classification approach.

## 5. Discussion

Recently, deep learning techniques such as convolutional networks and transfer pre-trained learning have shown significant success in many medical image analysis applications. To aid clinicians and professionals in predicting the early identification of breast cancer, we propose an artificial intelligence-based ETECADx framework based on the ensemble learning as well as the transformer-based self-attention mechanism to identify the breast cancer lesions for both binary and multi-class approaches. To achieve the goal of the study, six pre-trained individual deep learning models are adopted and used: DenseNet201, VGG16, GoogleNet, InceptionResNetV2, Xception, and ResNet50. In the proposed ETECADx model, we use the ensemble learning as a backbone network to aid the ViT in providing the superior classification performance.

### 5.1. The Findings of the Binary Approach

DenseNet201 and VGG16 achieve the best classification accuracy in pre-trained models for the binary approach recording 95.74% for both. The InceptionResNetV2 recorded a prediction accuracy of 93.62%, while the Res-Net50 model performed with the lowest accuracy achieving 85.11%. In contrast, the ensemble learning model outperformed the AI individual models in accuracy by 97.16%, whereas the proposed ETECADx model outperformed against the ensemble model with an accuracy of 98.58%. The performance in terms of F1-score is also recorded to be 97.21% and 98.58% for the ensemble learning model and the proposed hybrid ETECADx models, respectively. Furthermore, the ensemble learning model is recorded to be 0.8973, and 0.8961 for MCC and Kappa, respectively. Whereas the ETECADx model reaches better than the ensemble model with 0.9461 for MCC and 0.9461 for Kappa. The ensemble learning model is predicted three cases wrong for the normal class, and one case in the abnormal class. Meanwhile the ETECADx model could distinguish predictions better than the ensemble learning model with one case wrong only in both normal and abnormal classes.

Moreover, the proposed ensemble learning and ETECADx models are compared to other AI individual pre-trained models in terms of the number of fine-tune layers, train-able parameters, costs of training time consumption for each epoch, and testing time for each model and per image, as shown in Table 8. The InceptionResNetV2 model with fine-tune 672 and the DenseNet201 model with fine-tune 481 are the heaviest models, while GoogleNet and Xception are relatively light. The proposed ETECADx model is a combination of the three best AI models that have already been trained; it may still need more time cost, and epochs, but it provided a promising evaluation performance as well as outperforming the state-of-the-art AI individual models.

The standard deviation is also computed to evaluate the variety of the AI models with respect to the proposed ETECADx model. Figure 11 shows the results of this study performance characteristics for all AI models. It is clearly indicated that the Densenet201 and VGG16 had the closest accuracy performance compared with the proposed model, since the prediction error is 0.028. On the other hand, ResNet50 shows the worst error deviation in terms of SPE with 0.234 and 0.204 F1-scores. The closest model with ETECADx is the Xception model in terms of sensitivity and AUC, with error rates of 0.0277 and 0.0274, respectively. Furthermore, the Densenet201 model achieves the best match with the ETECADx, achieving the lowest specificity error deviation with 0.039. To determine whether the proposed ensemble ETECADx is significantly prominent, performing better predictions than other individual or even ensemble AI models, the paired *t*-test with a significance level of 0.05 *p*-value is investigated. Assuming the null hypothesis, there is no significant performance difference between our proposed models and others. Whereas, the alternative hypothesis is the opposite to show how much the proposed model is prominent. To perform the *t*-test study, we conduct two experiments using the evaluation metrics of the models in charge. First, the paired *t*-test is investigated between the best individual VGG16 and the ensemble AI models. The *p*-value of 0.11 is recorded to show that the ensemble model is more significant than the individual models. Second, we repeat the same study between the ensemble model against the proposed ETECADx where the *p*-value is recorded to be 0.08. Thus, the null hypothesis is rejected to show that the proposed ETECADx is significantly prominent than the ensemble as well as the individual models.

### 5.2. Multi-Classification Approach Findings

For the multi-class approach, the best evaluation accuracies of the pre-trained models are recorded to be 95.74%, 95,04%, and 94.33% via VGG16, Xception, and DenseNet201, respectively. Unfortunately, GoogleNet could not perform well; it achieves the lowest accuracy with 85.11%. As shown in Figure 7, the individual models predict the benign and malignant cases similarly, where three and two cases are misclassified, respectively. For normal cases prediction, the VGG16 performs better than other models, where it misclassifies only one case to be as a benign case. In addition, the proposed ensemble learning model (i.e., VGG16, Xception, and DenseNet201) could achieve better performance than individual models providing overall accuracy of 96.45%. Furthermore, the proposed ETECADx framework could outperform all individual or ensemble models achieving an overall accuracy of 97.87%. The performance in terms of F1-score is recorded to be 96.83% and 97.85% via the ensemble learning and ETECADx models, respectively. Moreover, the ensemble learning model is recorded to be 0.8683 and 0.8676 for MCC and Kappa, respectively. Whereas the ETECADx model performs better than the ensemble model with 0.9210 for MCC and 0.9206 for Kappa. As shown in Figure 10b, the proposed ETECADx could predict very well, as only one case is misclassified in terms of normal and malignant cases, achieving better rates of FPs and FNs. Generally, the ensemble learning model misclassified five cases (i.e., two in benign, one in malignant, and two in normal cases), while the ETECADx could predict very well, as only one case per class is misclassified. For comparing the proposed models to other AI models in multi-class, Table 9 shows the comparison performance in terms of number of fine-tune layers, trainable parameters, costs of training time consumption for each epoch, and testing time for a single breast image. As presented in Table 9, the DenseNet201 model is the heaviest model, while the GoogleNet has the lowest testing time cost.

For evaluating the variety of models with regard to the proposed ETECADx model, Figure 12 shows the standard deviation performance characteristics for all AI models in the multi-classification approach. The prediction error is 0.021 for the VGG16 model, which performed the closest accuracy performance when compared to the proposed model. With 0.317 in terms of SPE and 0.296 F1-score, InceptionResNetV2 demonstrates the worst error deviation. With error rates of 0.077, the Xception model has the best AUC compared to ETECADx. Additionally, the ETECADx model achieves the lowest specificity error deviation of 0.048 and the Densenet201 model has the best match with ETECADx. Similarly, we perform the paired *t*-test to determine the significance of the multi-class approach results. The paired samples test between the best pre-trained individual model (i.e., VGG16) and the ensemble model produce a *p*-value of 0.16, which is greater than the significance level of 0.05. In contrast, the *p*-value between the ensemble and ETECADx models is recorded to be 0.46. In this case, the null hypothesis is rejected and the alternative hypothesis shows the proposed ETECADx is significantly more prominent than individual and ensemble models.

### 5.3. Ablation Study Using the Real Breast Image Dataset

As described in Section 3.2, we used the private real dataset to validate and verify the capability of the proposed ETECADx framework ability in handling unseen mammograms. The real dataset contains 3D X-ray breast images with confidence labels annotated by three expert radiologists. It consists of 110 normal, 25 begin, and 101 malignant breast images. To perform the ablation study, we similarly apply both binary and multi-classification approaches. For the binary classification approach, the benign and malignant cases are combined together to represent the abnormal cases. For the multi-classification approach, the normal, benign, and malignant cases are used as independent classes.

#### 5.3.1. Approach A: Binary Classification Approach

The validation results for the binary classification approach of all AI models including the proposed ensemble and ETECADx are shown in Table 10. It is clearly shown that the proposed ETECADx model achieved the highest prediction performance in terms of all evaluation mercies: 97.16% Accuracy, 97.16% Sensitivity, 97.18% Specificity, 97.16% F1-score, and 97.19% AUC.

As shown in Figure 13 for the binary approach on real datasets, confusion matrices of the ensemble learning model misclassifies six abnormal and two normal cases. Meanwhile, the ETECADx model could predict the abnormal cases very well, as only two abnormal cases are wrongly predicted. Thus, the proposed ETECADx performs well, achieving a better performance in terms of FPs and FNs as well compared with the individual AI models or even with the ensemble model. To compare the evaluation performance of the proposed ETECADx using both INbreast and unseen real dataset, we designed a further investigation study to show the capability of the ETECADx to predict breast cancer from the unseen images during the training time. Figure 14 shows the evaluation of the binary approach for the ETECADx in contrast to an ensemble deep learning model using both INbreast and real datasets.

#### 5.3.2. Approach B: Multi-Class Classification Approach

For the multi-class approach, Table 11 presents the validation results of the proposed ETECADx framework against the ensemble learning model and other AI individual models using the real dataset (i.e., Dataset2-B). We can summarize that the proposed ETECADx model could successfully outperform other individual and their ensemble models. The best validation classification results are achieved via the proposed ETECADx model by recording 89.41% Accuracy, 89.41% Sensitivity, 91.27% Specificity, 87.77% F1-score, and 83.83% AUC. The ensemble learning model got twenty-one cases wrong predicted for benign cases, and six cases in malignant cases. Meanwhile, the ETECADx model is distinguished in prediction better than the ensemble learning model with wrong predictions of eighteen, one, and six cases in benign, malignant, and normal cases, respectively, as shown in Figure 15. Similarly, we perform one more validation study to validate the capability of the proposed ETECADx system predicting new mammogram instances. The proposed CAD system is trained only using INbreast images, while the real breast images are totally isolated during the training time. Figure 16 summarizes the validation results using both INbreast and real datasets for the multi-class approach.

### 5.4. Comparison Evaluation Results against the Latest Research Works

In Table 12, we present the results of the comparison among the proposed ETECADx and the most recent deep learning studies for breast cancer classification. It is possible that the proposed ETECADx could produce competitive and encouraging evaluation results on real-world datasets. For this study, we summarize the related studies that used INbreast dataset for indirect comparisons. Such indirect comparison lacks fair comparison with other studies in the literature research domain due to different dataset distribution, different data splitting settings, different AI models used, or even different execution environments to perform the AI models training and evaluation.

### 5.5. Limitations and Future Work

Since medical image labeling is expensive and time-consuming for radiologists, it is still the main limitation for the supervised-based AI applications. The automatic ROIs extraction is required for the practical applications where we intend to deploy such detectors for real application scenarios for hospitals and cancer centers in order to assist the physicians providing accurate and rapid diagnosis decisions. As we prove in our previous works [10], segmentation and detection of the suspicious ROIs could be performed automatically before the classification stage as a mandatory practical procedure of any CAD system. Using automatic segmentation or detection AI algorithms, such as MobileNetSSDv2, Detectron2, Faster-RCNN, or YOLO, could automate the proposed ETECADx for practical applications not just for breast cancer but for various medical imaging modalities in future studies. Moreover, we would test the proposed AI model again by collecting breast cancer risk factors from patients alongside images of their medical diagnoses (e.g., ultrasound or mammograms). In addition, we intend to integrate the newest outstanding AI technologies, such as explainable AI [67], and federated learning [68], to further enhance the performance behavior and provide more interesting breast cancer identification performances.

## 6. Conclusions

This article presents the potential of using hybrid AI models to build a novel ETECADx framework for breast cancer identification. We design and build the proposed CAD system by combining the recent emerging techniques of the ensemble learning of multiple AI-based CNN models and the Transformer encoder (i.e., ViT) to improve breast cancer prediction using digital X-ray mammograms. The INbreast public multi-class dataset is used to train and evaluate the proposed ETECADx framework. Meanwhile, private real breast cancer images are collected and annotated by expert radiologists to validate and verify the prediction performance. In this study, a comprehensive experimental study is similarly performed in terms of investigating the prediction performance for binary and multi-classification approaches in three stages: (1) individual pre-trained transfer learning; (2) ensemble deep learning; and (3) the proposed ETECADx model based on the ensemble Transformer encoder. For the binary approach, the evaluation results of the proposed deep learning ETECADx framework are recorded as 98.58% accuracy, a 97.31% F1-score, and 0.9461 for MCC. For the multi-class identification approach, the overall prediction performance achieved 97.87% accuracy, a 94.80% F1-score, and 0.9210 for MCC. For predicting a single mammogram, the proposed ETECADx shows its capability to identify the breast cancer type within an average of 0.048 s. Based on the evaluation metrics, the proposed model is superior and more effective than other state-of-the-art deep learning methods for detecting breast cancer in its earliest stages using digital X-ray breast images. Furthermore, the evaluation of private real datasets shows the proposed ETECADx model’s potential to yield reasonable accuracy results.

## Figures and Tables

**Figure 1 diagnostics-13-00089-f001:**
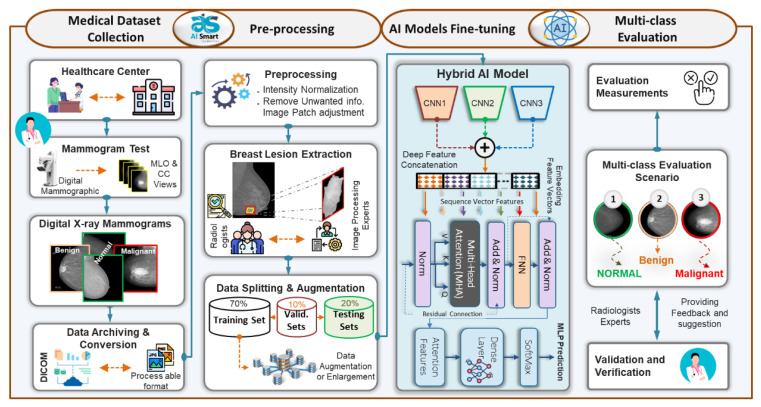
Abstract view of the proposed ETECADx framework to distinguish the breast cancer lesions to normal, benign, and malignant tissues.

**Figure 2 diagnostics-13-00089-f002:**
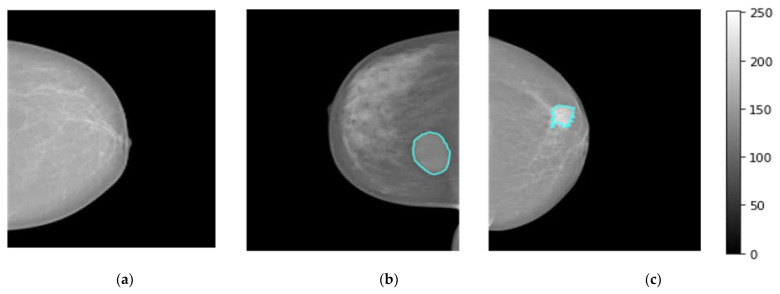
Examples of breast cancer mammograms from INbreast dataset (i.e., dataset1) for normal, benign, and malignant cases as depicted in (**a**–**c**), respectively. The breast tumor regions of interest (ROIs) for benign or malignant cases are accurately determined by expert radiologists as they are superimposed in a blue contour on the original mammogram.

**Figure 3 diagnostics-13-00089-f003:**
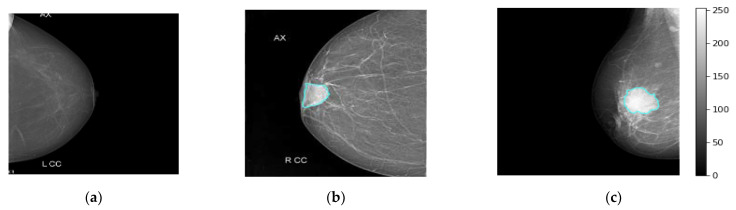
Some examples of real breast cancer mammograms. Normal, benign, and malignant cases are depicted in (**a**–**c**), respectively. The breast lesion localization is accurately superimposed on the breast mammograms of benign and malignant cases via the cyan color counter.

**Figure 4 diagnostics-13-00089-f004:**
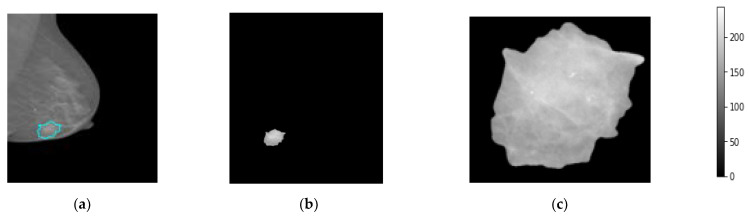
Patch image extraction to represent the breast region of interest, ROI. (**a**) Original mammogram, (**b**) the segmented lesion area, and (**c**) the final extracted ROI patch images.

**Figure 5 diagnostics-13-00089-f005:**
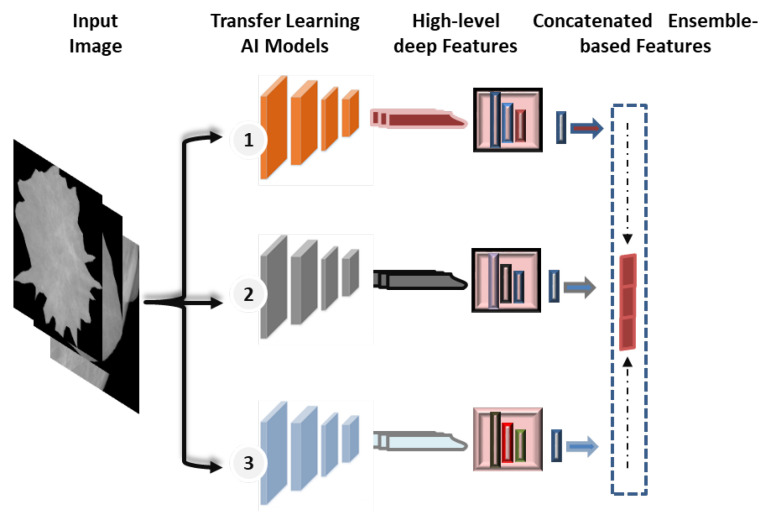
Concatenated ensemble-based features strategy.

**Figure 6 diagnostics-13-00089-f006:**
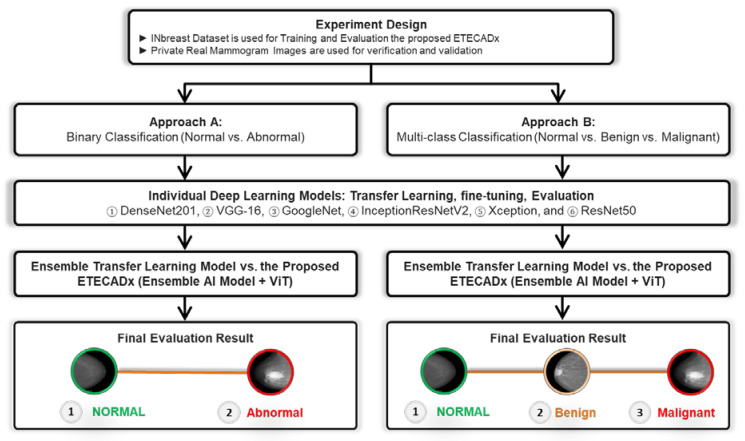
Experimental Study Scenario: *Approach A* and *Approach B*.

**Figure 7 diagnostics-13-00089-f007:**
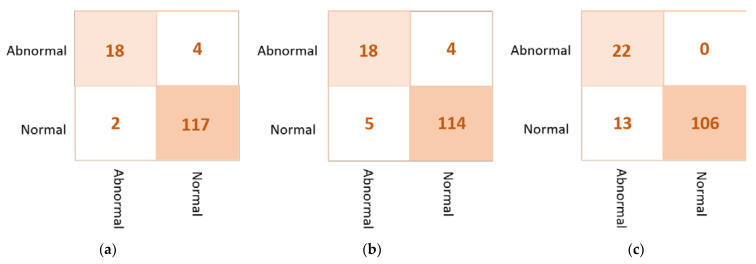
Example of the confusion matrices in the *binary approach* for individual pre-trained models using INbreast testing set: (**a**) DenseNet201, (**b**) InceptionResNetV2, and (**c**) Xception.

**Figure 8 diagnostics-13-00089-f008:**
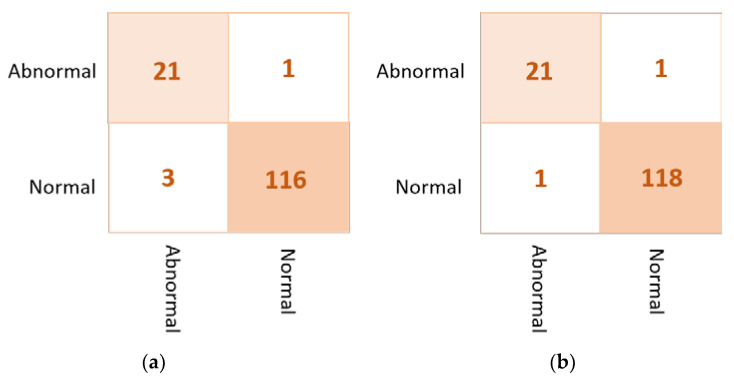
Evaluation performance in terms of the confusion matrices of the proposed AI models. (**a**) Ensemble learning as a backbone model and (**b**) the proposed AI-based ETECADx model.

**Figure 9 diagnostics-13-00089-f009:**
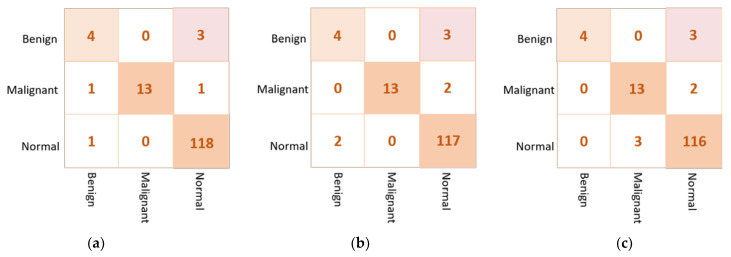
Confusion matrices of the best three individual models for the multi-classification approach: (**a**) VGG16, (**b**) Xception, and (**c**) DenseNet201.

**Figure 10 diagnostics-13-00089-f010:**
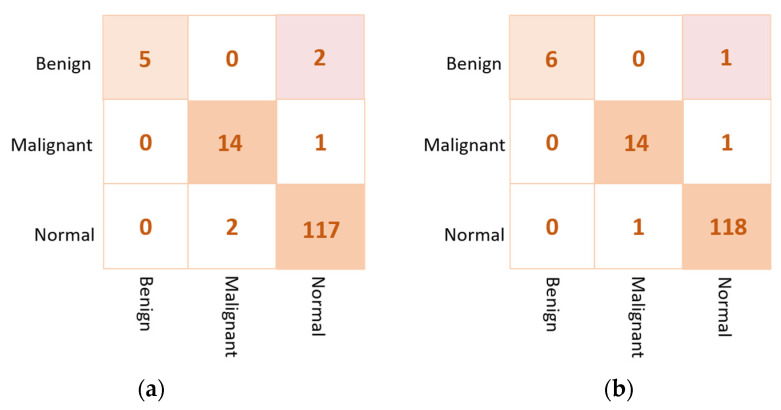
Confusion matrix of the proposed AI models for the multi-class classification approach. (**a**) The Ensemble learning model (**b**) the proposed ETECADx.

**Figure 11 diagnostics-13-00089-f011:**
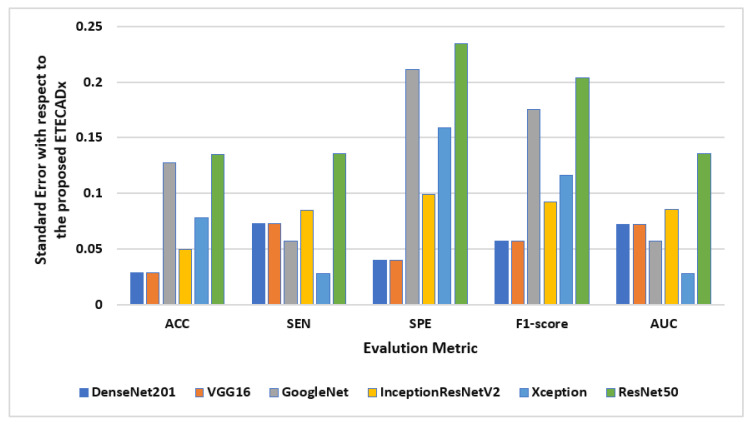
Evaluation error performance of the individual AI models with respect to the proposed ETECADx model for the binary approach.

**Figure 12 diagnostics-13-00089-f012:**
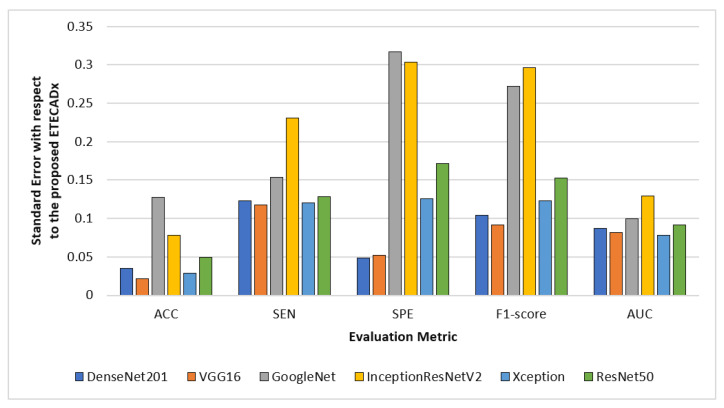
Evaluation error performance of the individual models with respect to the proposed ETECADx model in the multi-class approach.

**Figure 13 diagnostics-13-00089-f013:**
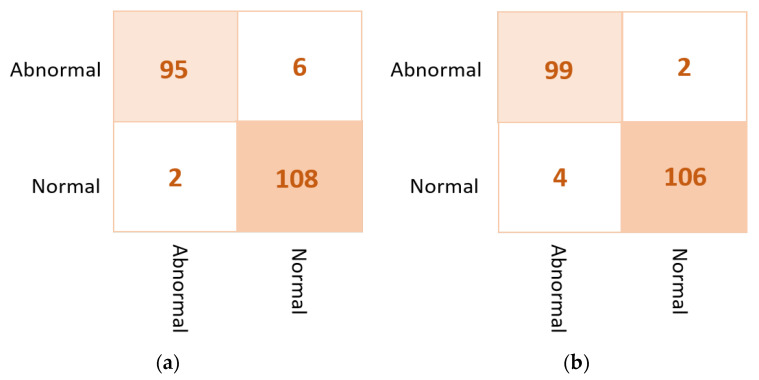
Confusion matrix of the proposed AI models for the binary classification approach on real dataset: (**a**) the ensemble learning model and (**b**) the proposed ETECADx.

**Figure 14 diagnostics-13-00089-f014:**
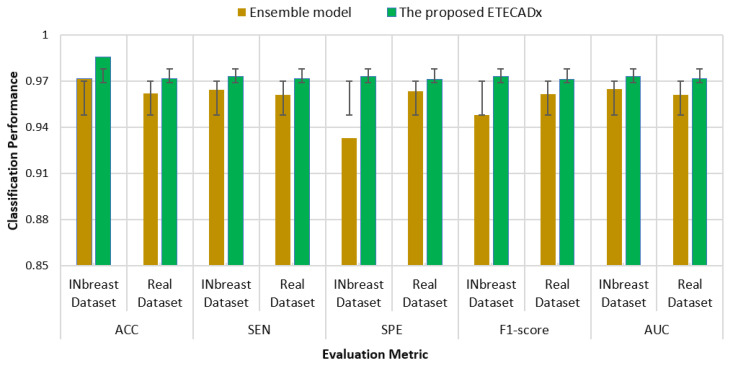
Comparison results of the binary approach for the ETECADx model against the ensemble learning model in both INbreast and real datasets.

**Figure 15 diagnostics-13-00089-f015:**
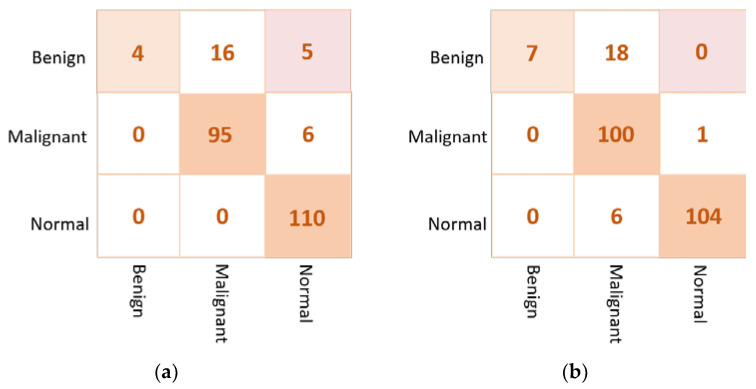
Confusion matrix of the proposed AI models for the multi-class classification approach on real dataset: (**a**) the Ensemble learning model (**b**) the proposed ETECADx.

**Figure 16 diagnostics-13-00089-f016:**
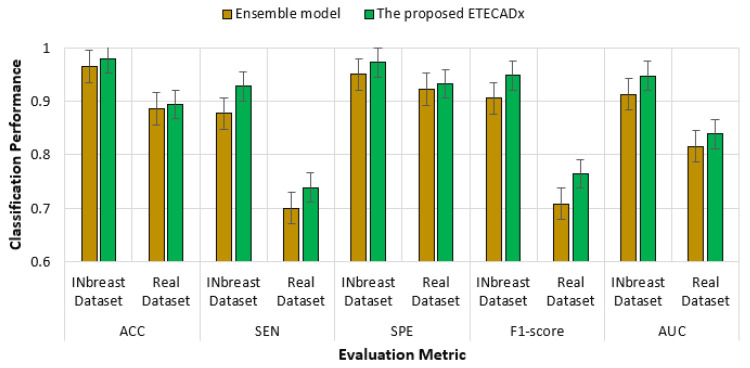
Comparison results of multi-class approach for ETECADx model against the ensemble learning model using both INbreast and real datasets.

**Table 1 diagnostics-13-00089-t001:** Real benchmark private breast images dataset. The class-wise labels (normal, benign, and malignant) were evaluated twice with three local and international expert radiologists.

Dataset	Evaluation Type	Annotation Type	Normal	Benign	Malignant
Dataset2-A	Internal evaluation: two local expert radiologists	Class-wise label	100	54	126
Dataset2-B *	External evaluation: international expert radiologist	Class-wise label and contour the breast tumor	100	25	101

* Dataset2-B is the final dataset that we use to validate and verify the proposed ETECADx framework.

**Table 2 diagnostics-13-00089-t002:** INbreast data distribution of Approach A (binary classification) of each class. The number of augmented training breast images is listed in the second column of each class.

Data Splitting	Normal	Abnormal	Total
Training (70%)	418	74	74
Training + Augmentation		592	1010
Validation (10%)	60	11	71
Testing (20%)	119	22	141
Total	597	625	1222

**Table 3 diagnostics-13-00089-t003:** INbreast data distribution of Approach B (multi-class classification) of each class. The number of augmented training breast images is listed in the second column of each class.

Data Splitting	Normal	Benign	Malignant	Total
Training (70%)	418	25	49	74
Training + Augmentation		400	392	1210
Validation (10%)	60	4	7	71
Testing (20%)	119	7	15	141
Total	597	411	414	1422

**Table 4 diagnostics-13-00089-t004:** Binary approach classification evaluation results of the individual pre-trained deep learning models using the testing set of the INbreast dataset.

AI Model	ACC	SEN	SPE	F1-Score	AUC	MCC	Kappa
DenseNet201	0.9574	0.9574	0.9565	0.9566	0.9007	0.8335	0.8322
VGG16	0.9574	0.9574	0.9565	0.9566	0.9007	0.8335	0.8322
GoogleNet	0.8582	0.8582	0.9257	0.8738	0.9160	0.6601	0.6070
InceptionResNetV2	0.9362	0.9362	0.9375	0.9360	0.8881	0.7623	0.7620
Xception	0.9078	0.9078	0.9420	0.9157	0.9459	0.7482	0.7178
ResNet50	0.8511	0.8511	0.8924	0.8637	0.8377	0.5672	0.5442

**Table 5 diagnostics-13-00089-t005:** Binary approach classification evaluation performance of the proposed ETECADx framework against the ensemble learning model using the testing set of INbreast dataset.

AI Model	ACC	SEN	SPE	F1-Score	AUC	MCC	Kappa
Ensemble learning model	0.9716	0.9716	0.9733	0.9721	0.9647	0.8973	0.8961
The proposed hybrid ETECADx: Ensemble + ViT	0.9858	0.9858	0.9858	0.9858	0.9731	0.9461	0.9461

**Table 6 diagnostics-13-00089-t006:** Approach B multi-classification evaluation results of the individual pre-trained deep learning models using the testing set of the INbreast dataset.

AI Model	ACC	SEN	SPE	F1-Score	AUC	MCC	Kappa
DenseNet201	0.9433	0.9433	0.9452	0.9412	0.8603	0.7855	0.7836
VGG16	0.9574	0.9574	0.9558	0.9549	0.8658	0.8365	0.8312
GoogleNet	0.8511	0.8511	0.9185	0.8766	0.8478	0.5963	0.5709
InceptionResNetV2	0.9007	0.9007	0.9183	0.9030	0.8179	0.6530	0.6478
Xception	0.9504	0.9504	0.9508	0.9500	0.8695	0.8128	0.8115
ResNet50	0.9291	0.9291	0.9370	0.9321	0.8552	0.7423	0.7418

**Table 7 diagnostics-13-00089-t007:** Evaluation performance of the proposed ETECADx framework against the ensemble learning model for the multi-class classification approach using the INbreast testing set.

AI Model	ACC%	SEN	SPE	F1-Score	AUC	MCC	Kappa
Ensemble learning model	0.9645	0.9645	0.9656	0.9638	0.9131	0.8683	0.8676
The proposed hybrid ETECADx: Ensemble + ViT	0.9787	0.9787	0.9788	0.9785	0.9506	0.9210	0.9206

**Table 8 diagnostics-13-00089-t008:** Performance comparison of the proposed ensemble learning and the hybrid ETECADx models against the individual AI models to the computation costs in binary approach.

AI Model	No. of Fine-Tune Layers	No. of Trainable Parameters (Million)	Training Time/Epoch (ms)	Testing Time/Image (s)	Frame Per Second (FPS)
DenseNet201	481	8.95	187	0.018	55.56
VGG16	17	2.89	197	0.024	41.67
GoogleNet	252	12.64	164	0.009	250
InceptionResNetV2	672	18.49	680	0.054	18.52
Xception	106	10.50	164	0.009	250
ResNet50	143	17.08	169	0.013	76.92
Ensemble model		32.06	238	0.047	21.28
The proposed ETECADx		25.74	260	0.048	20.83

**Table 9 diagnostics-13-00089-t009:** Performance comparison of the proposed ensemble and ETECADx models against other individual AI models for the multi-classification approach.

AI Model	No. of Fine-Tune Layers	No. of Trainable Parameters (Million)	Training Time/Epoch (ms)	Testing Time/Image (s)	Frame Per Second (FPS)
DenseNet201	481	08.95	195	0.018	55.56
VGG16	17	02.890	186	0.02	50
GoogleNet	252	12.644	165	0.008	125
InceptionResNetV2	720	12.384	174	0.012	83.33
Xception	96	12.119	150	0.010	100
ResNet50	123	19.316	182	0.011	90.91
Ensemble model		25.491	270	0.047	21.28
The proposed hybrid ETECADx: Ensemble + ViT		19.571	290	0.048	20.83

**Table 10 diagnostics-13-00089-t010:** Validation classification results for the *binary approach* using all AI models including the ensemble learning and the proposed ETECADx framework. The real unseen dataset (Dataset2-B) is used for this study.

AI Model	ACC	SEN	SPE	F1-Score	AUC	MCC	Kappa
DenseNet201	0.9005	0.9005	0.9164	0.8990	0.8965	0.8155	0.7988
VGG16	0.9526	0.9526	0.9566	0.9524	0.9505	0.9087	0.9046
GoogleNet	0.7725	0.7725	0.8388	0.7635	0.7814	0.6116	0.5525
InceptionResNetV2	0.7820	0.7820	0.8502	0.7734	0.7909	0.6322	0.5711
Xception	0.9289	0.9289	0.9323	0.9289	0.9310	0.8612	0.8580
ResNet50	0.9573	0.9573	0.9595	0.9574	0.9587	0.9167	0.9147
Ensemble model	0.9621	0.9621	0.9627	0.9619	0.9620	0.9245	0.9239
The proposed hybrid ETECADx: Ensemble + ViT	0.9716	0.9716	0.9718	0.9716	0.9719	0.9432	0.9430

**Table 11 diagnostics-13-00089-t011:** Validation classification results for the *multi-classification approach* using all AI models including the ensemble learning and the proposed ETECADx framework. The real unseen dataset (Dataset2-B) is used for this study.

AI Model	ACC	SEN	SPE	F1-Score	AUC	MCC	Kappa
DenseNet201	0.8771	0.6543	0.5850	0.6177	0.7899	0.7921	0.7774
VGG16	0.8690	0.7798	0.7409	0.7478	0.8621	0.7831	0.7783
GoogleNet	0.6229	0.5741	0.6897	0.5652	0.6812	0.4347	0.3620
InceptionResNetV2	0.6186	0.5278	0.7072	0.5554	0.6476	0.3343	0.3166
Xception	0.8771	0.6543	0.5850	0.6177	0.7899	0.7847	0.7829
ResNet50	0.8686	0.7184	0.7293	0.7217	0.8267	0.7762	0.7744
Ensemble model	0.886	0.886	0.896	0.857	0.8158	0.8052	0.7949
The proposed hybrid ETECADx: Ensemble + ViT	0.8941	0.8941	0.9127	0.8777	0.9071	0.8252	0.8123

**Table 12 diagnostics-13-00089-t012:** Comparison evaluation results against the latest AI research works for breast cancer.

Reference	Dataset	Classes	Prediction Method	Accuracy (%)
Samee et al. (2022), [3]	INbreast	Normal/Abnormal	AlexNet, VGG, and GoogleNet	98.50
Al-antari et al. (2018), [10]	INbreast	Benign/Malignant	YOLOV2	95.32
Chakravarthy et al. (2022), [33]	INbreast	Normal/Abnormal	ICSELM	98.26
Hamed et al. (2020), [29]	INbreast	Benign/Malignant	YOLO model	89.50
Hamed et al. (2021), [24]	INbreast	Benign/Malignant	YOLOV4	95.0
Aly et al. (2021), [22]	INbreast	Benign/Malignant	YOLO v3	89.40
Chakravarthy et al. (2022), [18]	INbreast	Benign, Malignant, and Normal	CNNs and SVM	96.64
Shen et al. (2019), [20]	INbreast	5-class (Normal, Benign, Malignant with mass and calcification)	Many CNN models	86.70 (SEN) 96.10 (SPE)
Lee, Sanghoon, et al. (2019), [62]	The Cancer Genome Atlas (TCGA),	Tumor vs. lymphocytes	Ensemble of SVM, LR, and RF	96.90
Kadam et al. (2019), [63]	UCI WDBC dataset	Benign and Malignant	Ensemble of Sparse Autoencoders and Softmax Regression	98.60
Moon et al. (2020), [64]	Private Ultrasound dataset	Benign, Malignant, and Normal	Ensemble of VGGNet, ResNet, and DenseNet	94.62
Abbasniya et al. (2022), [65]	BreakHis	Benign and Malignant	Inception-ResNet-v2, Ensemble of (CatBoost), (XGBoost) and (LightGBM)	LightGBM has given the best average accuracy
Jiang et al. (2022), [66]	CBIS-DD, INbreast, MIAS	Benign and Malignant(mass/calcification)	PAA, EfficientNet-B3 and two-stage detector + image classifier	96.30 (INbreast)89.40 (CBIS-DDSM)
He, Zhu, et al. (2022), [43]	BreakHis dataset	Benign and Malignant	Deconv-Transformer (DecT)	93.02
The proposed ETECADx Framework: Hybrid ensemble learning and ViT	INbreast & private real mammogram dataset	Approach A: Normal and Abnormal	The proposed AI Hybrid Model (Ensemble backbone and ViT)	98.58
		Approach B: Normal, Benign, and Malignant		97.87

## Data Availability

The datasets used in this paper are publicly available at: https://www.kaggle.com/datasets/ramanathansp20/inbreast-dataset (accessed on 15 November 2022). The implementation source code of this study is available online via https://github.com/AymenMuslihAlHejri/AI_Breast_Cancer (accessed on 27 November 2022).

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
