# Peer review of "ETECADx: Ensemble Self-Attention Transformer Encoder for Breast Cancer Diagnosis Using Full-Field Digital X-ray Breast Images"

_diagnostics, 2022, doi:10.3390/diagnostics13010089_

Round 1
Reviewer 1 Report
The overall structure of the paper is appropriate, and the manuscript was written quite well. Some comments are provided to improve the quality of this manuscript. In my opinion, this manuscript should be revised before accepting in Diagnostics.
1) Abstract is too long, please consider summarize it (200-250 words).
2) There are some grammatical errors throughout the paper. Below are some of the most eye-catching ones
a) " VGG16, DenseNet201, ResNet50, transformers (ViT), ensemble, and son"
b) " Ours Model "
3) Ref [3] is mentioned twice in "Related Works" section, please consider removing one of them.
" Samee et al.[3] used AlexNet, VGG, and GoogleNet models to extract fea-128 tures from the INbreast dataset, while such extracted features were minimized by involv-129 ing univariate approach, reaching 98.50%,98.06%,98.99%, 98.98% for accuracy, sensitivity"
" Finally, the authors in [3] used AlexNet, VGG, and GoogleNet 142 for feature extraction, while the dimension of the extracted features was reduced using a 143 univariate methods. The proposed model achieved 98.50% accuracy, 98.98% sensitivity, 144 98.99% specificity, and 98.06% precision"
4) Some of the references in the "Related Works" section are outdated ([10], [11], [24]). It is highly recommended to replace them with the following papers, which are most recent papers on breast cancer diagnosis.
[1] M.R. Abbasniya, S.A. Sheikholeslamzadeh, H. Nasiri, S. Emami, Classification of Breast Tumors Based on Histopathology Images Using Deep Features and Ensemble of Gradient Boosting Methods, Comput. Electr. Eng. 103 (2022) 108382.
[2] J. Jiang, J. Peng, C. Hu, W. Jian, X. Wang, W. Liu, Breast cancer detection and classification in mammogram using a three-stage deep learning framework based on PAA algorithm, Artif. Intell. Med. 134 (2022) 102419. 4) Algorithm 1 and Algorithm 2 are highly dependent on OpenCV, please consider rewrite them in general form and independent of a specific library or framework.
5) Please bold the best results in each Table.
6) The test set is imbalanced (Based on Fig. 6 and Fig. 7), I recommend to use methods like Synthetic Minority Oversampling Technique (SMOTE) to oversample the minority class.
7) Please consider adding a column to Table 13 and mention the Dataset used in each reference or number of samples in each class.
8) Please provide more discussion on the results.
9) No statistical tests were performed in the paper. So, how could we determine whether the results are statistically significant?
Author Response
First, we would like to thank reviewer#1 for his/her insightful comments and suggestions to improve the overall manuscript's innovation and structure.
We have addressed and answered the reviewer’s comments point-by-point. Accordingly, the manuscript is revised based on the reviewer's suggestions and recommendations as highlighted in the revised manuscript.
Thanks!

Reviewer 2 Report
1. Paper is well written.
2. Authors have provided experimental results for single models and ensemble models. It is expected that the authors clearly show the way transfer learning is used (detailed architectures of customized layers) in their work.
3. It is further expected that authors should show how ensemble learning is used as the backbone (for eg which method of ensemble learning is used? the way models are combined).
4. It is suggested to show performance comparison with other work where ensemble learning is used rather than a single model.
5. Authors should provide limitations of existing work and the scope of improvements.
6. It is expected that authors should show graphs of each model for training and validation accuracy as well as training and validation loss.
7. I suggest increasing the number of studies and adding a new discussion there to show the advantage. The following studies can be considered
1. A Bottom-Up Review of Image Analysis Methods for Suspicious Region Detection in Mammograms
2. Image Augmentation Techniques for Mammogram Analysis
Author Response
Dear Editors in the Diagnostic Journal
Manuscript Information: Title: “ETECADx: Ensemble Self-Attention Transformer Encoder for Breast Cancer Diagnosis Using Full-Field Digital X-Ray Breast Images”ID: diagnostics-2097648
First, we would like to thank reviewer#2 for his/her insightful comments and suggestions to improve the overall manuscript's innovation and structure.
We have addressed and answered the reviewer’s comments point-by-point. Accordingly, the manuscript is revised based on the reviewer's suggestions and recommendations as highlighted in the revised manuscript.
Thanks!

Reviewer 3 Report
up to this line is not needed in the abstract.
{Breast cancer is one of the most frequent harmful diseases in women's societies world- 22 wide. It is the most prevalent cancer and has silent growing-up characteristics to attack and 23 damage the normal breast tissues.}
The section 1 needs expansion. figure 9 and 10 are similar in what way they are different?
section 4 needs clarification. Conclusion may be modified.
Time complexity analysis may be included. MCC,FM and Kappa are to be calculated.
Author Response
Dear Editors in the Diagnostic Journal
Manuscript Information: Title: “ETECADx: Ensemble Self-Attention Transformer Encoder for Breast Cancer Diagnosis Using Full-Field Digital X-Ray Breast Images”ID: diagnostics-2097648
First, we would like to thank reviewer#3 for his/her insightful comments and suggestions to improve the overall manuscript's innovation and structure.
We have addressed and answered the reviewer’s comments point-by-point. Accordingly, the manuscript is revised based on the reviewer's suggestions and recommendations as highlighted in the revised manuscript.
Thanks!

Reviewer 4 Report
Breast cancer is the most leading cancer occurring in women and is a significant factor in female mortality. Early diagnosis of breast cancer with Artificial Intelligent (AI) developments for breast cancer detection can lead to a proper treatment to affected patients as early as possible that eventually help reduce the women mortality rate. In this paper the authors proposed AI-based computer-aided diagnosis (CAD) framework called ETECADx by fusing the benefits of both ensemble transfer learning of the convolutional neural networks as well as the self-attention mechanism of vision transformer encoder (ViT). Using real dataset they compared the proposed system with existing models.
1 English can be improved. Proofreading should verify that grammar, tense, and punctuation are used correctly.
The length of the paper may be reduced.
Author Response
Dear Editors in the Diagnostic Journal
Manuscript Information: Title: “ETECADx: Ensemble Self-Attention Transformer Encoder for Breast Cancer Diagnosis Using Full-Field Digital X-Ray Breast Images”ID: diagnostics-2097648
First, we would like to thank reviewer#4 for his/her insightful comments and suggestions to improve the overall manuscript's innovation and structure.
We have addressed and answered the reviewer’s comments point-by-point. Accordingly, the manuscript is revised based on the reviewer's suggestions and recommendations as highlighted in the revised manuscript.
Thanks!

Round 2
Reviewer 1 Report
In general, the authors have reflected the comments of reviewers quite well and careful. In my opinion, the idea and experiments are qualified for publishing on Diagnostics. Some comments are provided to improve the quality of this manuscript:
1) In Table 13, please consider replacing the word "Year" with year of publication of each reference for the following references:
"Shen et al., year is missing."
"Brunese et al. (Year)"
"Kadam et al. (Year)"
"Moon et al (Year)"
"Abbasniya et al. (Year)"
"Jiang ea al. (2022), [Ref#]" , reference number is missing.
"Zhu et al. (Year)"
2) Please complete details of Ref [51].
"OpenCV: Image Thresholding," 2022.
Author Response
First, we would like to thank reviewer#1 for his/her round 2 comments and suggestions to improve the manuscript.
Thanks!

Reviewer 3 Report
All the corrections are Included in the paper and hence the paper may be accepted.
Author Response
First, we would like to thank reviewer#3 for his/her round 2 support the manuscript.
Thanks!